# Plasmonic layer-selective all-optical switching of magnetization with nanometer resolution

D.O. Ignatyeva [1,2,6]*, C.S. Davies [3,4,6]*, D.A. Sylgacheva [1,2], A. Tsukamoto[5], H. Yoshikawa[5], P.O. Kapralov[2], A. Kirilyuk[4], V.I. Belotelov[1,2] & A.V. Kimel[3]

All-optical magnetization reversal with femtosecond laser pulses facilitates the fastest and least dissipative magnetic recording, but writing magnetic bits with spatial resolution better than the wavelength of light has so far been seen as a major challenge. Here, we demonstrate that a single femtosecond laser pulse of wavelength 800 nm can be used to toggle the magnetization exclusively within one of two 10-nm thick magnetic nanolayers, separated by just 80 nm, without affecting the other one. The choice of the addressed layer is enabled by the excitation of a plasmon-polariton at a targeted interface of the nanostructure, and realized merely by rotating the polarization-axis of the linearly-polarized ultrashort optical pulse by 90°. Our results unveil a robust tool that can be deployed to reliably switch magnetization in targeted nanolayers of heterostructures, and paves the way to increasing the storage density of opto-magnetic recording by a factor of at least 2.

[1] Faculty of Physics, Lomonosov Moscow State University, 119991 Moscow, Russia. [2] Russian Quantum Center, 45 Skolkovskoye Shosse, 121353 Moscow, Russia. [3] Radboud University, Institute for Molecules and Materials, 135 Heyendaalseweg, 6525 AJ Nijmegen, The Netherlands. [4] FELIX Laboratory, Radboud University, 7 Toernooiveld, 6525 ED Nijmegen, The Netherlands. [5] College of Science and Technology, Nihon University, 7-24-1 Funabashi, Chiba 274-8501, Japan. [6] These authors contributed equally: D. O. Ignatyeva, C. S. Davies. *email: daria.ignatyeva@gmail.com; c.davies@science.ru.nl

Over the course of more than half a century, local magnetic fields delivered by recording heads in hard-disk drives have successfully driven writing speeds towards gigahertz frequencies and storage densities beyond 1 Tb/in$^2$. However, this progress is now becoming restricted by the onset of the superparamagnetic limit[1], and so new approaches that facilitate faster and denser magnetic recording technologies are now being sought[2]. The process of single-shot all-optical switching[3], whereby magnetization can be switched deterministically using just an ultrashort optical pulse of arbitrary polarization, fundamentally promises near-terahertz writing speeds[4,5] in ferrimagnetic alloys of gadolinium-iron-cobalt (GdFeCo). In essence, all-optical switching within GdFeCo originates from the ultrafast magnetization dynamics[6,7] of the exchange-coupled Gd and FeCo sublattices, triggered by an ultrafast transfer of thermal energy from the light to the spin bath. This process strongly depends on the amount of energy delivered by the optical pulse[8]. If the incident fluence is below the minimum threshold required for all-optical switching, only ultrafast demagnetization[9] with subsequent recovery of the net magnetization is achieved. On the other hand, if too much energy is supplied by the pulse, a random distribution of switched and unswitched magnetic domains are formed in the irradiated region.

A variety of effects can mediate the transfer of energy between the incident optical pulse and the illuminated material. The material property of magnetic circular dichroism or the optical property of interference, for example, can tune the efficiency of energy transfer. These effects and properties can be respectively employed to achieve helicity-dependent all-optical switching[8] or to reduce the incident optical fluence needed for all-optical switching by two-thirds[10]. While optical pulses trigger thermally-driven magnetization dynamics in metallic GdFeCo systems[8], it is also possible to non-thermally generate effective magnetic fields using circularly-[11] or linearly-polarized[12–14] femtosecond pulses in alternative materials. In addition, upon illuminating a dielectric/metallic interface with p-polarized light, a surface plasmon polariton (SPP) can be excited[15]. The optical field enhancement associated with SPPs could offer a pathway towards an even smaller energy cost of all-optically writing a bit[13,14,16]. At the same time, by confining the optical field using a plasmonic antenna, the optically-addressed region of magnetization has been successfully downscaled to length scales on the order of 50 nm[17].

Supplementary to reducing the in-plane size of the optically-addressed region of magnetization, an intriguing possibility for achieving higher storage densities lies in extending magnetic platters in to the third dimension. Recently, substantial research has been directed towards all-optically reversing magnetization in multi-layered heterostructures, motivated by the prospect of stacking multiple magnetic layers on the same platter[18,19] or constructing optically-addressable magnetic tunnel junctions[20]. Using magnetic bilayers, one can already achieve ultrafast single-shot switching of Co/Pt (via exchange-coupling with a Gd[21] or GdFeCo[22] layer), and realize a form of multi-level magnetic recording based on the generation of spin-polarized currents[23]. If one, however, considers a simple bilayer of GdFeCo optically-addressed at normal incidence, the directionality of the optical wave vector suggests—at first glance—that the first layer addressed will always and inescapably undergo switching. Until now, no method has been demonstrated that allows for switching in the second layer, without affecting the first.

In this article, we unveil a method of deterministic layer-resolved all-optical switching in a multi-layered heterostructure of GdFeCo, exploiting the properties of SPPs. Using calculations, we design the thermal distributions generated by s- and p-polarized light across the depth of a heterostructure. Upon exposing the

bilayer to an s-polarized ultrashort optical pulse, no SPP is generated, and so the energy is concentrated at the first incident layer penetrating only the skin depth. On the contrary, using a p-polarized ultrashort optical pulse, we generate an SPP that concentrates energy in the second layer (the GdFeCo/air interface) rather than the first GdFeCo/glass interface. We experimentally verify this scenario, and moreover exploit the different concentrations of energy in order to all-optically switch the magnetization of targeted nanolayers of GdFeCo independently from each other, using p- and s-polarized ultrashort optical pulses. We therefore demonstrate an approach of polarization-dependent multi-level magnetic recording, whereby each magnetic layer can be independently addressed at will, thus enabling the storage density of optically-addressed magnetic media to be doubled.

## Results

**Surface plasmon-polaritons in a multi-layered GdFeCo heterostructure.** Although the imaginary part of the dielectric permittivity of GdFeCo at the wavelength of 800 nm is large, the real part of the permittivity is negative and so the ferromagnetic metal can support the excitation of an SPP ($\varepsilon_{GdFeCo} = -5.91 + 19.17i$)[24]. At a semi-infinite air/GdFeCo interface, the transverse propagation length[15] and penetration depth inside the GdFeCo film[15] of an SPP is 2.6 μm and ∼17 nm respectively. To facilitate the excitation of an SPP, we have fabricated and studied the magnetic multi-layered heterostructure prism/glass/Si$_3$N$_4$(5)/Gd$_{26.0}$Fe$_{64.8}$Co$_{9.2}$(10)/Si$_3$N$_4$(80)/Gd$_{27.0}$Fe$_{63.9}$Co$_{9.1}$(10)/Si$_3$N$_4$(10) (Fig. 1a), where the number in parentheses indicates the layer thickness in nanometres. The ferrimagnetic GdFeCo layers were deliberately designed to have different concentrations of gadolinium, and will be discussed in detail later on. The glass substrate is coupled directly to a 60° SiO$_2$ prism, allowing for an SPP to be optically excited via the Kretschmann scheme[15] at the bottom GdFeCo/air interface. The thin (10 nm- and 5 nm-thick) Si$_3$N$_4$ layers merely protect the GdFeCo from oxidation, and have negligible impact on the electromagnetic field distribution. On the other hand, the 80 nm-thick Si$_3$N$_4$ layer blocks the exchange interaction and diminishes the magneto-dipole interaction between the GdFeCo layers, ensuring their magnetization states are effectively decoupled. The 80 nm-thick Si$_3$N$_4$ layer also plays an integral role in allowing an incident p- and s-polarized pulses at the wavelength of 800 nm to deliver drastically different electromagnetic field distributions across the two GdFeCo layers (in general, this difference could be observed for Si$_3$N$_4$ layers with thickness up to 120 nm).

In order to demonstrate this effect, we model the optical absorption inside each GdFeCo layer by

$$A = \int \delta A(z)\mathrm{d}z = \frac{k_0}{n_{pr}\cos(\theta)} \int |e(z)|^2 \mathrm{Im}(\varepsilon_{GdFeCo})\mathrm{d}z, \quad (1)$$

where $n_{pr}$ is the refractive index of the coupling prism, $k_0$ is the optical wavenumber in vacuum, $\theta$ is the angle of incidence, $\delta A(z)$ is the partial absorption, and $e(z)$ is the normalized electric field (see "Methods"). Figure 1b, c show the calculated distribution of $|e(z)|^2$ and $\delta A(z)$ across the depth of the studied heterostructure for p- and s-polarized light at the wavelength of 800 nm incident on the top Gd$_{26}$(FeCo)$_{74}$ layer at an angle of 59°. The p-polarized light excites an SPP that is localized at the bottom Gd$_{27}$(FeCo)$_{73}$ layer (i.e. the GdFeCo/air interface), predominantly pumping the energy of the laser pulse into this layer. The calculations (Fig. 1e) show a strong dependence of $|e|^2$ at the bottom interface of the Gd$_{27}$(FeCo)$_{73}$ layer on the angle of incidence, with a maximum at 56°. We note that this is a characteristic feature associated with the Kretschmann geometry[25], and does not correspond to the

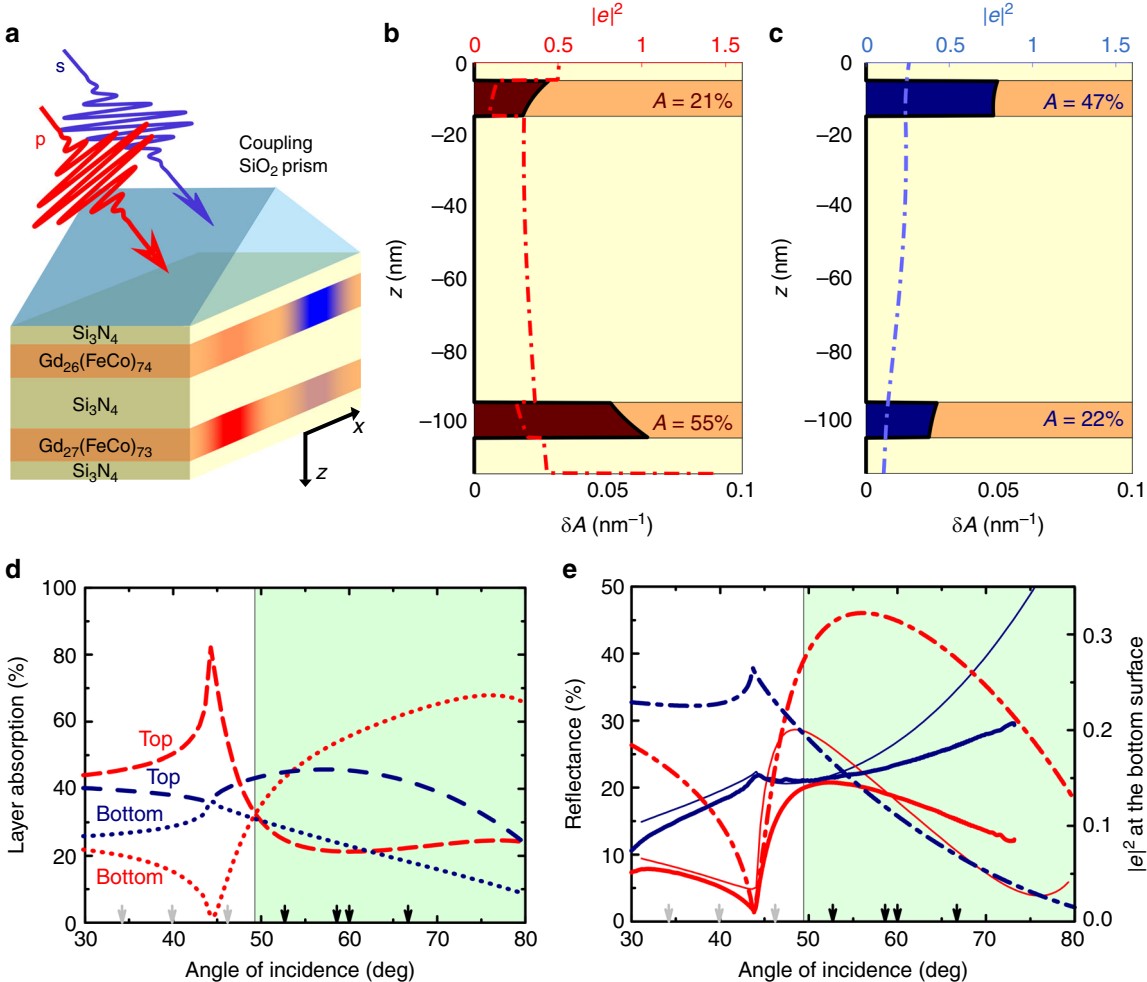

**Fig. 1** The scheme of polarization-based layer-resolved all-optical toggling of magnetization. **a** Physical scheme and sample design. The p-polarized/s-polarized optical pulse (and the associated targeted layer for magnetization reversal) is shown by red/blue. **b, c** The distribution of $|e|^2$ (dashed-dotted line) and the partial $\delta A$ and total $A$ absorption (solid line) inside the multi-layered heterostructure for p-polarized and s-polarized optical pulses respectively, with fixed wavelength 800 nm and angle of incidence 59°. **d** Calculated angular dependence of the absorbed optical energy by the top (dashed line)/bottom (dotted line) layer with incident p-polarized (red)/s-polarized (blue) light. **e** Calculated (thin solid lines) and experimentally measured (thick solid lines) reflectance angular spectra for incident p-polarized (red)/s-polarized (blue) light. Also shown is the calculated strength of $|e|^2$ at the bottom surface of the bottom $Gd_{27}(FeCo)_{73}$ layer with incident p-polarized (red dashed-dotted line)/s-polarized (blue dashed-dotted line) light. In panels d-e, the green background colour indicates the predicted region where the absorption of p-polarized light in the bottom $Gd_{27}(FeCo)_{73}$ layers is greater than in the top $Gd_{26}(FeCo)_{74}$ layer. This is the region with expected polarization-controlled layer selective switching of magnetization. The black and grey arrows indicate the angles where the polarization-based all-optical switching experimentally succeeded and failed respectively

maximum of the absorption in this layer. If the incident light is instead s-polarized, no SPP is excited, and therefore the light energy is partly reflected or partly dissipated in the top $Gd_{26}(FeCo)_{74}$ layer.

To tune the efficiency of SPP excitation[15], one can tune the material refractive index, the photon energy or the angle of incidence of light. We therefore numerically consider the former parameters to be fixed (mimicking experimental conditions), and calculate the angular dependence of the maximum absorbed energy in the top $Gd_{26}(FeCo)_{74}$ and bottom $Gd_{27}(FeCo)_{73}$ layers for incident p- and s-polarized light. Presented in Fig. 1d are the corresponding results, revealing that the SPP resonance is rather wide. Nevertheless, the graphs clearly show that if the angle of incidence is above ~50°, the absorption of p-polarized light in the bottom $Gd_{27}(FeCo)_{73}$ layer is greater than in the top $Gd_{26}(FeCo)_{74}$ layer. This represents the necessary condition for layer-selective magnetic recording and the region that satisfies this condition is shaded green in Fig. 1d. As a result of this

absorption difference, one may tune the pulse fluence such that the energy dissipated in the two magnetic layers is below or above the threshold required for the switching[8]. Hence layer selective switching with subwavelength resolution can be achieved. To verify this predicted angular dependence of absorption, we measured and calculated the angle-resolved reflectivity of p- and s-polarized light (Fig. 1e). Excellent qualitative and rather good quantitative agreement between the computationally predicted and experimentally observed angle dependencies evidences the high quality of the fabricated multi-layers and inspires our experimental search for layer selective all-optical magnetic recording in the heterostructures.

**Resolving the magnetization of different layers**. In order to distinguish the four different states of the magnetizations of the two $Gd_x(FeCo)_{100-x}$ layers with the help of a conventional magneto-optical microscope, the magnetic layers were

deliberately designed to have slightly different concentrations of gadolinium, with $x = 26$ ($x = 27$) in the top (bottom) layer. This resulted in a significant difference in the magnetization compensation temperatures $T_M$ ($T_M \approx 210$ K and $T_M \approx 315$ K for $Gd_{26}(FeCo)_{74}$ and $Gd_{27}(FeCo)_{73}$ respectively)[26,27]. Consequently, the magnetizations[28] as well as the coercive fields and the magneto-optical Faraday rotation of the layers at room temperature were also different. Both magnetic layers exhibit deterministic all-optical switching, with nearly the same optical fluence[29]. Since the values of $T_M$ lie on either side of room temperature, the net magnetization of the $Gd_{26}(FeCo)_{74}$ and $Gd_{27}(FeCo)_{73}$ layers is antiparallel and parallel respectively to the magnetization of the constituent Gd sublattice. The difference in coercive fields makes it possible to obtain four stable hysteretic states. After applying and removing an out-of-plane spatially-uniform magnetic field $H_B$, the four magnetic states of the heterostructure can be realized with mutually parallel (↑↑ and ↓↓) or antiparallel (↑↓ and ↓↑) alignment of the magnetizations in the layers[28,30]. In the notation adopted here, the first and second arrow denotes the polarity of magnetization of gadolinium in the top and bottom layer of GdFeCo respectively.

To evaluate whether deterministic layer-resolved all-optical switching can be achieved, we employ static imaging of the heterostructure using the polar magneto-optical Kerr effect. Subsequent to all tests of all-optical switching, we recorded magneto-optical images of the multi-layered system as a function of $H_B$. In Fig. 2b are shown raw images recorded for different values of $H_B$. The different magneto-optical signals from the two

layers allow us to directly distinguish the magnetic state of each layer, via the different contrast levels (which is mainly proportional to the out-of-plane component of magnetization $M_z$ of iron[31]. Moreover, by averaging all pixel intensities within the magneto-optical images, we are able to generate a characteristic hysteresis loop, as presented in Fig. 2a. This represents a superposition of the hysteresis loops belonging to both layers[27,30]. We use this loop to conclusively discern the four different magnetic states that can be achieved in our binary heterostructure.

**Layer-selective magnetization reversal.** To test the scenario of plasmon-enabled deterministic all-optical switching, we focused a 100 fs-long optical pulse, of central wavelength 800 nm, through the coupling prism at an angle of incidence of 59° onto the surface of the top $Gd_{26}(FeCo)_{74}$ layer (see Methods and Supplementary Note 1 for details). It follows from Fig. 1b that absorption of the p-polarized pulse at 59° in the bottom layer is about 2.5 times larger than in the top layer. At the same time, for these conditions the absorption of the s-polarized pulse in the top layer is about 2.2 times larger than in the bottom layer. Therefore, at this angle of incidence one should expect the layer-selective switching. The diameter of the focused spot was characterized[32] to be 37 μm. The magnetization of gadolinium within the heterostructure was initially set to the configuration ↓↑ (Fig. 2), and the linear polarization of the optical pulse was rotated to achieve p- or s-polarization. After exposing the sample to optical pulses, magneto-optical images were recorded and identified with one of the four states (↑↑, ↓↓, ↑↓ or ↓↑) using the values of the magneto-optical contrast in different parts of the hysteresis loop.

Exactly as predicted, we observed layer-selective single-shot switching of magnetization, with the choice of addressed layer dictated by the polarization of the optical pulse. Figure 3a shows a background-corrected magneto-optical image taken after the multi-layered heterostructure was exposed to a single p-polarized (left) and s-polarized (right) optical pulse, of fluence ~10 mJ/cm² (electric field $|E| \sim 720$ MV/m). There is a clear difference in contrast between the two irradiated regions, indicating different magnetic states have been obtained. To irrefutably identify which layer(s) have undergone switching, we present also a cross-section taken from the magneto-optical image. By comparing this cross-section to those recorded in the presence of varying $H_B$ (Fig. 2), we conclusively demonstrate that the magnetization in the bottom $Gd_{27}(FeCo)_{73}$ and top $Gd_{26}(FeCo)_{74}$ layer has been independently toggled by the p-polarized and s-polarized optical pulse respectively. This layer-resolved magnetization reversal effect was observed for all four different starting magnetic states (see Supplementary Note 3), and the deterministic switching of magnetization in different layers could be achieved by delivering several optical pulses (see Supplementary Note 4). Finally, upon removing the coupling prism, we were unable to achieve layer-resolved all-optical switching, verifying further the critical role the SPP plays in this mechanism.

As discussed earlier, the all-optical switching of magnetization in a single layer can only be achieved if the density of the absorbed optical fluence $w$ exceeds a minimum threshold value $w_{threshold}$[22], which is almost identical in both layers. Therefore, for a certain incident optical fluence $F$, the density of the absorbed fluence $w_j = A_j F$ in layer $j$ depends only on the angle of incidence and polarization (which enter the absorption term, $A_j$). If one increases the optical fluence, it is conceivable that one could achieve all-optical switching in both layers, since the fluence threshold in both layers can be simultaneously surpassed (i.e., $w_{top,bottom} > w_{threshold}$). Upon therefore increasing the optical fluence to ~12 mJ/cm² ($E \sim 790$ MV/m), as shown in Fig. 3b, we observe a new feature with different contrast at the centre of the

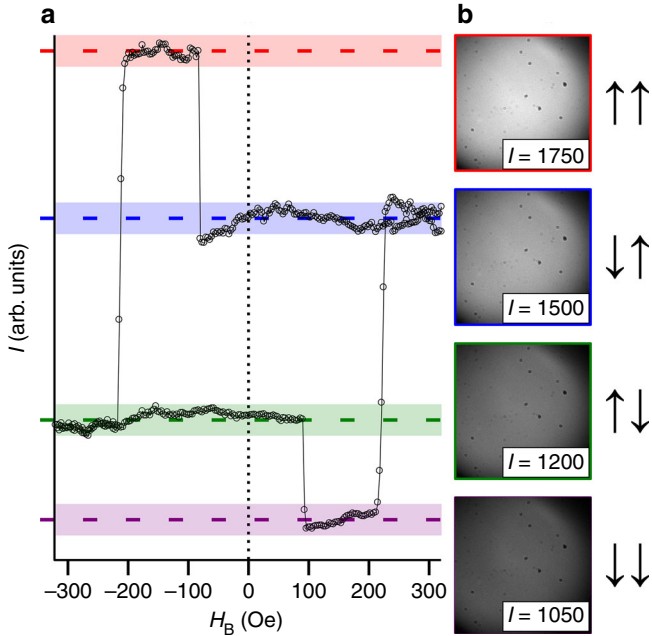

**Fig. 2** Characteristic hysteresis loop of the multi-layered heterostructure obtained with the help of magneto-optical microscopy. The magneto-optical signal $I$ is proportional to the out-of-plane component of the magnetization of the iron sublattice in the entire heterostructure. **a** Characteristic hysteresis loop of the entire heterostructure, constructed by averaging the pixel intensities of magneto-optical images recorded at varying strengths of the out-of-plane bias magnetic field $H_B$. **b** Typical raw magneto-optical images, measured with $H_B = 0$, and the inset value corresponds to the average pixel intensity of the associated image. The four shaded levels of the hysteresis loop correspond to the four stable magnetic states that can be obtained, with the left (right) arrow indicating the orientation of the magnetization of gadolinium within the top (bottom) layer of GdFeCo

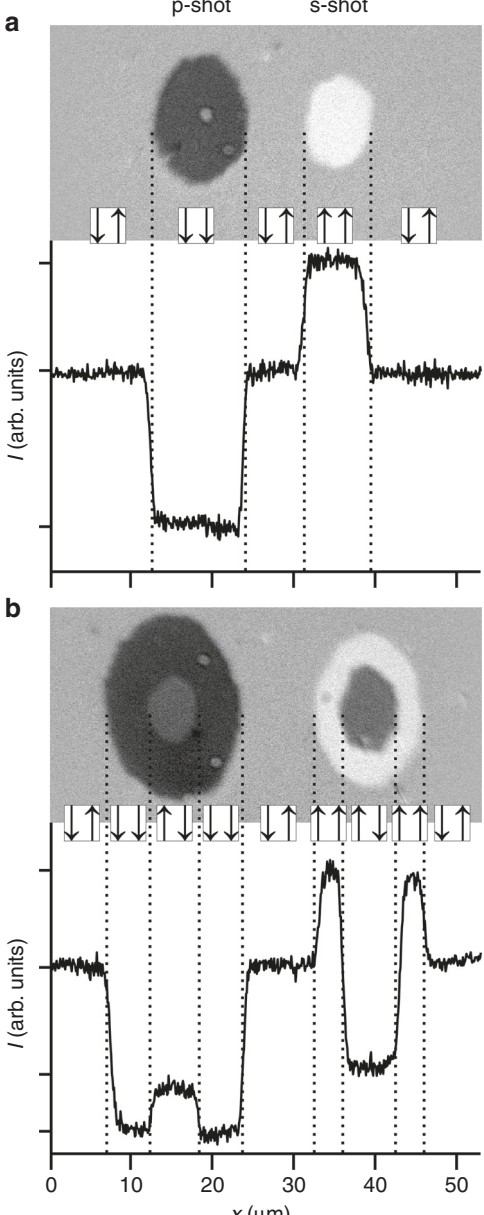

**Fig. 3** Polarization-dependent all-optical switching of magnetization in different layers of the heterostructure. Background-corrected magneto-optical images are shown, taken after exposing the heterostructure to a single p-polarized (left spot) and s-polarized (right spot) optical pulse, incident at an angle of 59°. Also shown is a cross-section (averaged over a width of 1.8 μm) extracted from the image. The left (right) arrow indicates the orientation of the magnetization of Gd within the top (bottom) layer of GdFeCo. The results shown in **a** and **b** were obtained with an incident fluence of ~10 mJ/cm$^2$ and ~12 mJ/cm$^2$ respectively

irradiated region. Due to the Gaussian distribution of the optical pulse energy, it is expected that there is a higher thermal load supplied at the centre. Quantitative analysis (bottom panel of Fig. 3b) reveals that, at the centre, both layers have indeed undergone all-optical switching, irrespective of the optical polarization. These observations not only reveal an important prerequisite for layer-resolved all-optical switching in terms of the laser fluence, but also shows that by tuning both the polarization and fluence of the laser pulses, one can realize all four states of the magnetizations of the layers (↑↑, ↓↓, ↑↓ and ↓↑).

The crucial role of the plasmon-polariton in the layer selective magnetic switching is further emphasized by experiments on optical control of the magnetizations in the multi-layer structure performed at different angles of incidence. The model presented in Fig. 1d–e predicts that angles of incidence above ~50° will allow a p-polarized optical pulse to switch magnetization in the bottom GdFeCo layer only. We therefore varied the angle of incidence between 34° and 67°, and the switching of the bottom layer was only seen within the range of 53° and 67°. If instead the angle was tuned between 34° and 47°, the top Gd$_{26}$(FeCo)$_{74}$ layer always underwent switching, irrespective of the optical polarization. All magneto-optical images underpinning this conclusion are supplied in Supplementary Note 5. The black arrows in Fig. 1d–e indicate the angles at which plasmon-enabled switching was observed, and the grey arrows indicate the opposite, demonstrating excellent agreement of the experimental results with the model. This finding not only proves the important role of the SPP in the demonstrated mechanism, but also uncovers the robustness of the effect, since the reversal is unaffected despite adjusting the angle of incidence by at least 30°.

## Discussion

In conclusion, we have revealed an elegant solution to the problem of layer-selective all-optical magnetic recording in multi-layered heterostructures. Our approach is based on the layer selective deposition of optical energy through the polarization-dependent excitation of a surface plasmon-polariton at a targeted interface of the heterostructure. The numerically-predicted and experimentally-verified fact that the polarization-selective magnetization reversal occurs only in the vicinity of the surface plasmon-polariton's resonance (and is otherwise absent) clearly and conclusively shows the key role played by the surface plasmon-polaritons in the process of layer-resolved all-optical reversal of magnetization.

While, for the sake of simplicity, we have restricted ourselves to studying a multi-layered heterostructure containing only two magnetic layers, it is straightforward to incorporate more magnetic layers. To suitably address these different layers individually without affecting the others, one needs to adjust not only the polarization-axis of the incident optical pulse but also the wavelength and/or the angle of incidence[15]. More intrinsically, each GdFeCo layer could be neighboured by different dielectric layers with specifically designed refractive indices. This could assist or obstruct the process of switching within the particular layer for a certain optical pulse. Indeed, since we do not need to tune the energy of the optical pulse to switch either of the two GdFeCo layers studied here, the ultimate recording density is limited not by the number/thickness of magnetic layers but rather by the number of distinct appropriately-tailored energy distributions achievable across the multi-layered stack using different optical parameters. The dielectric prism may also be replaced with an etched or transient[33] grating, with the SPP resonance position controlled by the grating periodicity. In this way, a complete multi-level magnetic recording architecture could be developed, with different optical parameters dictating the choice of which magnetic layer is addressed.

## Methods

**Numerical calculations.** Partial and full absorption inside the lossy planar structure, in the case of the oblique optical incidence, is calculated using the energy conservation law

$$W = S_z^{inc} \cdot s + S_z^{t} \cdot s + S_z^{r} \cdot s, \qquad (2)$$

where $S_z$ is the z-component of the Poynting vector of the incident (inc), transmitted (t) or reflected (r) light, and $s$ is the surface area parallel to the xy-plane. $W$ is the energy absorbed by the volume that, according to the Joule-Lenz law,

depends on the conductivity $\sigma$ via

$$W = \frac{1}{2} \int \sigma |E|^2 s \mathrm{dz}. \tag{3}$$

Taking in to account $A = 1 - R - T$, where $A$, $R$ and $T$ correspond to the absorption, reflection and transmission coefficients respectively, Eq. (2) can be rearranged to obtain

$$\frac{W}{S_z^{\mathrm{inc}}} = 1 - R - T, \tag{4}$$

Combining Eqs. (2) and (3) yields

$$A = \frac{1}{2} \frac{\int \sigma |E|^2 s \mathrm{dz}}{S_z^{\mathrm{inc}}}. \tag{5}$$

Using the identities $\sigma = \frac{\omega \mathrm{Im}[\varepsilon]}{4\pi}$ and $S_z^{\mathrm{inc}} = \frac{c n_{\mathrm{prism}}}{8\pi} |E_{\mathrm{inc}}|^2 \cos\theta$, where $\omega$ and $c$ are the frequency and speed of light, and $\varepsilon$ is the permittivity, one obtains Eq. 1, i.e., $A = \int \delta A(z) \mathrm{dz} = \frac{k_0}{n_{\mathrm{prism}}\cos(\theta)} \int |e(z)|^2 \mathrm{Im}(\varepsilon) \mathrm{dz}$. Here, $e(z)$ is the electric field normalized to the value of the electric field for the incident light, i.e., $e(z) = E(z)/|E_{\mathrm{inc}}|$, and the magnitude of the electric field is given by $|E_{\mathrm{inc}}| = \sqrt{2I/c\varepsilon_0 n_{\mathrm{prism}}}$.

**Sample fabrication**. For the experimental study, we prepared multi-layered heterostructure deposited by magnetron sputtering on an atomically flat glass substrate. The ferrimagnetic GdFeCo nanolayers were fabricated by co-sputtering of Gd, Fe, and Co elements with the direct current (DC) magnetron sputtering method. As an advantage of amorphous alloy systems, we can tune concentration of alloy continuously by controlling the relative deposition rates and can precisely design the coercive field at room temperature as shown in Fig. 2. The thin $Si_3N_4$ layers were fabricated by reactive radio frequency (RF) magnetron sputtering of Si with nitrogen gas. Calibration of deposition rate and optical characterization were performed by spectroscopic ellipsometry. Thin film with high refractive index (~2.00 at wavelength of 800 nm) was obtained on the optimal deposition conditions.

**Experimental setup for measuring the reflectance angular spectra**. The reflectance angular spectra at the pump wavelength was measured using a rotating motorized platform STANDA 8MR174-11. Incident light was generated by a single-mode diode laser THORLABS L785P090, with a spectral line width of about 0.1 nm. The temperature of the laser case was stabilized with an accuracy of 0.02 °C. A parallel light beam was formed using an aspheric lens with a focal length of 6 mm. Then the light passed through a film polarizer, an optical chopper modulating light at a frequency of about 500 Hz, and a diaphragm. The intensity of the reflected light was measured using a THORLABS FDS1010 silicon photodiode connected to a low-noise transimpedance amplifier. The signal from the amplifier was digitized by a National Instruments USB-6351 data acquisition board.

**Experimental setup for achieving and detecting polarization-dependent toggle-switching**. To achieve polarization-dependent all-optical switching of magnetization, we used an amplified Ti:Sapphire laser system to generate ~100 fs optical pulses, with a central wavelength of 800 nm and a Gaussian spatial distribution. The optical pulses either had a repetition rate of 1 kHz (used when aligning or measuring the power) or were triggered in single-shot mode (when achieving all-optical switching). A polarizer and half-wave plate were used to rotate the polarization-axis of the linearly polarized pulse, and the pulse was then routed towards the coupling prism, and focussed using a lens of focal length 25 cm. The angle of incidence (relative to the coupling prism) was measured by taking and analysing a photograph of two diaphragms centred on the laser beam.

Detection of the magnetic state of the multi-layered heterostructure was achieved using a magneto-optical microscope. A Euromex 100 W halogen LE.5210 lamp supplied illumination, which was linearly polarized by a polarizing sheet and then directed through an objective lens (Mitsutoyo, G Plan APO × 20, NA = 0.28). Upon reflection, the polarization-axis of the illuminating light was rotated due to the polar magneto-optical Kerr effect. The reflected beam was then spatially separated using a non-polarizing beam splitter, and passed through a Glan-Taylor polarizer (the latter was almost crossed with the former polarizing film). Finally, a spatially-resolved image (with an amplitude proportional to the out-of-plane component of magnetization) was recorded using a CCD (QImaging, Retiga R3) coupled to a variable magnifier. All images were recorded with an exposure time of 250 ms. A short-pass filter, with a cut-off wavelength of 750 nm, was used to prevent the 800 nm optical pulse from polluting the CCD. An electromagnetic pole was positioned directly behind the coupling prism in order to supply an out-of-plane magnetic field (maximum attainable strength ± 320 Oe).

## Data availability
The data that support the plots within this paper and other findings of this study are available from the corresponding authors Daria Ignatyeva (daria.ignatyeva@gmail.com) or Carl Davies (c.davies@science.ru.nl) upon reasonable request.

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

## Acknowledgements
The authors thank S. Semin and C. Berkhout for technical support, and K. Mishra for support during the experiments. This research has received funding from Russian Foundation for Basic Research, grant No. 18-32-20225, from the European Union's Horizon 2020 research and innovation program under FET-Open Grant Agreement No. 713481 (SPICE) and FET-Open Grant Agreement No. 737093 (FEMTOTERABYTE), de Nederlandse Organisatie voor Wetenschappelijk Onderzoek (NWO), and from Grant-in-Aid for Scientific Research on Innovative Area,"Nano Spin Conversion Science", grant No. 26103004. The authors thank the Erasmus + program between Radboud University and Lomonosov Moscow State University. D.A.S. acknowledges support from BASIS Foundation scholarship.

## Author contributions
D.O.I and V.I.B conceived and designed the experiments, and D.O.I performed the numerical calculations. A.T. and H.Y. fabricated the multi-layered structure. P.O.K. measured the angle-resolved reflectance. C.S.D. and D.A.S. performed the experimental measurements of layer-resolved all-optical switching, and C.S.D. processed the experimentally-collected results. D.O.I. and C.S.D. co-wrote the manuscript with contributions from V.I.B., A.V.K., and A.K. The work was coordinated by A.V.K.

## Competing interests
The authors declare no competing interests.
