## [Peer Review File · Nature Communications]

Reviewers' comments:

Reviewer #1 (Remarks to the Author):

The authors demonstrated layer-selective all-optical magnetic recording in multi-layered heterostructures. The proposed approach is based on the polarization-dependent excitation of a surface plasmon-polaritons in a the Kretschmann configuration and it can be expanded to a grating coupling scheme. The layer-selective all-optical magnetic recording can be observed in a wide range of the incidence angle, at least 30 degrees. To my knowledge this study presents the first experimental demonstration of the layer-selective all-optical magnetic recording and would pave the way to all-optical multilevel 3D magnetic recording. I feel that the results presented are of immediate interest to others in the community and people from several disciplines. I recommend publishing the paper after some improvements. A few suggestions and questions are summarized below.

-The excitation of SPP in the GdFeCo/air interface can induce an effective magnetic field in the transverse y-direction due to the plasmon-induced IFE [Song-Jin Im et al., Physical Review B 96, 165437 (2017), Song-Jin Im et al., Physical Review B 99, 041401(R) (2019)]. It is easy to investigate from the simulation results magnitude of the plasmon-induced IFE in comparison with the out-of-plane bias magnetic field H_B (Fig. 2). Discussion on influences of the plasmon-induced IFE or at least mentioning why this effect is disregarded is required to make a thorough study.

-In the simulation results (Fig. 1(b) and (c)) the electric field would be better represented in V/m, not in the arb. unit, which is directly achievable from the simulation results and is more informative for readers to estimate magnitude of the electric field.

- The model for the optical absorption in the line 107 should be improved so that it can directly provide $A(\%)$ and $\Delta A(\text{nm}^{-1})$ in Fig. 1. The well-known formula of absorption density is $J \cdot E = \omega \cdot (1/2) \cdot \text{abs}(E)^2 \cdot \text{Im}(\epsilon)$ in the unit of W/m^3 .

-The authors claimed the robustness in relation with the incident angle. Investigating a robustness in relation with the layer thickness parameters are also important for the fabrication process and to determine the magnetic recording density. Are the separation layer thickness 80 nm and the GdFeCo layer thickness 10 nm the minimum for the highest density, or are they optimized for a certain purpose?

Reviewer #2 (Remarks to the Author):

This study demonstrates an interesting combination of ultrafast laser pulse driven plasmonics and all-optical magnetic writing. A GdFeCo double layer system is used as a magnetic medium, separated by a dielectric. Plasmons can be coupled into the films via a prism, to match the wave vectors and then deposit the heat. Via the laser polarization, the layers can be chosen where the plasmons are selectively excited. In the GdFeCo material system, the magnetization can be switched by heating via toggle switching. This is successfully demonstrated.

It is a very nice idea that is a combination of different areas of physics, thus of interest for a broad readership. The idea could be potentially downscaled, since plasmonic receivers can be built very small and are used to squeeze the light waves today. I support a publication with my full endorsement.

Reviewer #3 (Remarks to the Author):

In the paper "Plasmonic layer-selective all-optical switching of magnetization with nanometer resolution," the authors report on selective all-optical magnetization reversal in two 10nm thin GdFeCo layers by changing the polarization of the incident fs pulse. Moreover, they claim that such selectivity due to the pulse polarization is born out of the generation of surface plasmon polaritons in the multilayer stack. The experiments are sound and the results reported by the authors

establish a path to achieve optical manipulation of magnetic bits beyond the diffraction limit of light. However, I would request the authors to clarify the following points before the work is accepted for publication:

1. The presence of a surface plasmon in the Kretschmann configuration is usually signified by a dip in the reflection spectra. In the reflection spectra provided in Fig 1e, two reflection dips around 45° and 75° angle of incidence can be seen for p-polarised light. However the authors chose to operate at 59° angle of incidence where no features are present in the reflection spectra. While the choice of angle definitely allows the authors to demonstrate polarization selective switching of magnetic layers due to different optical absorption, one can question whether that selectivity is due to the generation of surface plasmon polaritons or not. For example, can it not be the case that at 59° incidence, electromagnetic wave interference inside the multilayer stack creates different absorption within the two magnetic layers for s and p-polarised light. A more convincing case for the surface plasmon assisted switching can be made if the authors show the surface plasmon polariton dispersion curve along with a light line in SiO_2 . An intersection of the light line with the dispersion curve at 59° incidence would clearly indicate the generation of surface plasmons. Otherwise, one can say that the results are not due to surface plasmons but interference in the multilayer stack for two different light polarizations.

2. In Fig. 2, the curve for the magneto-optical intensity is for the Fe sublattice whereas the arrows are for the magnetization of the Gd atoms. Moreover it is also stated that in one layer the Gd atom magnetization are parallel and in the other layer it is anti-parallel to the net magnetization. In my opinion, it is a little confusing for readers who might not be so familiar with the intricacies of rare-earth ferromagnetic films. If the authors agree, wouldn't it make more sense to atleast change the arrows to reflect the net magnetization, since we are talking about switching magnetic layers?

Response to reviewers

We thank the reviewers for their positive assessment of our manuscript. Here, we address point-by-point all the comments provided by the reviewers. The text marked in blue corresponds to the comments from the reviewers, and the subsequent black text indicates our response. In the revised manuscript / supplementary material, the revisions are marked in red.

Reviewer #1 (Remarks to the Author):

The authors demonstrated layer-selective all-optical magnetic recording in multi-layered heterostructures. The proposed approach is based on the polarization-dependent excitation of a surface plasmon-polaritons in a the Kretschmann configuration and it can be expanded to a grating coupling scheme. The layer-selective all-optical magnetic recording can be observed in a wide range of the incidence angle, at least 30 degrees. To my knowledge this study presents the first experimental demonstration of the layer-selective all-optical magnetic recording and would pave the way to all-optical multilevel 3D magnetic recording. I feel that the results presented are of immediate interest to others in the community and people from several disciplines. I recommend publishing the paper after some improvements. A few suggestions and questions are summarized below.

We thank the reviewer for the positive evaluation of our work. Below, we address the suggestions and questions.

-The excitation of SPP in the GdFeCo/air interface can induce an effective magnetic field in the transverse y-direction due to the plasmon-induced IFE [Song-Jin Im et al., Physical Review B 96, 165437 (2017), Song-Jin Im et al., Physical Review B 99, 041401(R) (2019)]. It is easy to investigate from the simulation results magnitude of the plasmon-induced IFE in comparison with the out-of-plane bias magnetic field H_B (Fig. 2). Discussion on influences of the plasmon-induced IFE or at least mentioning why this effect is disregarded is required to make a thorough study.

The impact of a femtosecond pulse on a GdFeCo/air interface was investigated in Ref. [8], and it was shown that the non-thermal IFE plays a negligible role in comparison with thermal excitation of the spin system. Therefore, in contrast to dielectrics where optomagnetic effects can arise (e.g. from the effective magnetic field generated by circularly-polarized pulses, linearly-polarized pulses with oblique incidence, or SPPs), these effects are not observed in metallic GdFeCo films. We clarified this point by adding a sentence on p.2 of the revised paper, and added additional suitable references to [Song-Jin Im *et al.* Phys. Rev. B 96, 165437 (2017)] and [Song-Jin Im *et al.* Phys. Rev. B 99, 041401(R) (2019)].

-In the simulation results (Fig. 1(b) and (c)) the electric field would be better represented in V/m, not in the arb. unit, which is directly achievable from the simulation results and is more informative for readers to estimate magnitude of the electric field.

We agree with the reviewer that the magnitude of the electric field is an important quantity to discuss. So, we have added the magnitude of the electric field \mathbf{E} experimentally used to obtain the results shown in Fig. 3 (see p.9 and p.11 in the revised manuscript).

However, we would like to underline that the numerical results presented in Fig. 1 (b) and Fig. 1 (c) are, in general, better presented by normalizing to the incident electric field, rather

than quoted in V/m for some particular incident energy. In the numerical calculations, we assume the distribution of the electric field is top-hat in spatial distribution, whereas it is spatially-Gaussian in the experiment. Moreover, by normalizing to the incident electric field, we are able to numerically obtain the percent of the absorbed energy from the pulse directly (see comments to the next remark and the section discussing Numerical Methods on p.12). For these reasons the electric field in Fig. 1 (b)-(c) was normalized to the incident light.

In order to therefore avoid confusion, we have denoted the numerically-calculated normalized value of the electric field by $e(z)$ instead of $E(z)$.

- The model for the optical absorption in the line 107 should be improved so that it can directly provide $A(\%)$ and $\Delta A(\text{nm}^{-1})$ in Fig. 1. The well-known formula of absorption density is $J \cdot E = \omega \cdot (1/2) \cdot \text{abs}(E)^2 \cdot \text{Im}(\epsilon)$ in the unit of W/m^3 .

The formula in line 107 provides the absorption in % and nm^{-1} directly, and is derived from the Joule-Lenz formula and energy conservation law. We added details on the origin of this equation in Numerical Methods (p.12). Here, $|e|^2$ is normalized on its magnitude in the prism, which also corresponds to the normalization used in Fig. 1 (b)-(c).

-The authors claimed the robustness in relation with the incident angle. Investigating a robustness in relation with the layer thickness parameters are also important for the fabrication process and to determine the magnetic recording density. Are the separation layer thickness 80 nm and the GdFeCo layer thickness 10 nm the minimum for the highest density, or are they optimized for a certain purpose?

We have added information in the manuscript on how we selected the layer thickness for our proof-of-principle demonstration.

For the effect of polarization-based layer-targeted all-optical switching to work, it is compulsory for the GdFeCo layer to have a maximum thickness of ~ 20 nm. If the thickness of the metallic film exceeds this, the top layer of GdFeCo will absorb the majority of the optically-delivered energy, and prevent energy from reaching the layers below. In our article, we used a thickness of 10 nm since this thickness has been used extensively before in all-optical switching experiments. While it has been shown[R1] that 3 nm-thick GdFeCo layers feature PMA, the ability of ultrathin GdFeCo to display all-optical switching remains (to the best of our knowledge) untested. Nevertheless, since we did not need to change the optical energy to switch either of the two GdFeCo layers, we conclude that the optical parameters allow us to bypass the traditionally-accepted penetration depth of 11.0 nm in GdFeCo for $\lambda = 800$ nm[R2]. Thus, the maximum recording density is not limited by the thickness of GdFeCo, but rather by the number of tailored energy distributions that can be achieved across the multi-layered stack for different optical parameters.

With regards to the thickness of the Si_3N_4 layer, from the point of view of polarization-based layer-targeted switching using the same laser fluence, our calculations showed that the thickness could range between 25 nm and 120 nm. For 0-25 nm layer thickness, polarization-dependent switching is possible, but requires substantial change in the pulse energies for p- and s-polarized light due to differences in the Fresnel conditions. These low thicknesses also lead to undesirable exchange and magneto-dipolar coupling between the GdFeCo layers. For layers thicker than 120 nm, absorption of p-polarized light at the top layer grows, and prevents

polarization-based switching. The thickness of 80 nm produces the most symmetrical absorption distribution for p- and s-polarized light, and hence we selected this for the sample fabrication.

[R1] C.-M. Lee *et al.* IEEE Trans. Magn. **45**, 3808 (2009).

[R2] H. Yoshikawa *et al.* Jpn. J. Appl. Phys. **55**, 07MD01 (2016).

Reviewer #2 (Remarks to the Author):

This study demonstrates an interesting combination of ultrafast laser pulse driven plasmonics and all-optical magnetic writing. A GdFeCo double layer system is used as a magnetic medium, separated by a dielectric. Plasmons can be coupled into the films via a prism, to match the wave vectors and then deposit the heat. Via the laser polarization, the layers can be chosen where the plasmons are selectively excited. In the GdFeCo material system, the magnetization can be switched by heating via toggle switching. This is successfully demonstrated.

It is a very nice idea that is a combination of different areas of physics, thus of interest for a broad readership. The idea could be potentially downscaled, since plasmonic receivers can be built very small and are used to squeeze the light waves today. I support a publication with my full endorsement.

We thank the reviewer for the very encouraging comments about our work.

Reviewer #3 (Remarks to the Author):

In the paper “Plasmonic layer-selective all-optical switching of magnetization with nanometer resolution,” the authors report on selective all-optical magnetization reversal in two 10nm thin GdFeCo layers by changing the polarization of the incident fs pulse. Moreover, they claim that such selectivity due to the pulse polarization is born out of the generation of surface plasmon polaritons in the multilayer stack. The experiments are sound and the results reported by the authors establish a path to achieve optical manipulation of magnetic bits beyond the diffraction limit of light. However, I would request the authors to clarify the following points before the work is accepted for publication:

We thank the reviewer for the supportive evaluation of our manuscript.

1. The presence of a surface plasmon in the Kretschmann configuration is usually signified by a dip in the reflection spectra. In the reflection spectra provided in Fig 1e, two reflection dips around 45° and 75° angle of incidence can be seen for p-polarised light. However the authors chose to operate at 59° angle of incidence where no features are present in the reflection spectra. While the choice of angle definitely allows the authors to demonstrate polarization selective switching of magnetic layers due to different optical absorption, one can question whether that selectivity is due to the generation of surface plasmon polaritons or not. For example, can it not be the case that at 59° incidence, electromagnetic wave interference inside the multilayer stack creates different absorption within the two magnetic layers for s and p-polarised light. A more convincing case for the surface plasmon assisted switching can be made if the authors show the surface plasmon polariton dispersion curve along with a light line in SiO₂. An intersection of the light line with the dispersion curve at 59° incidence would clearly indicate the generation of surface plasmons. Otherwise, one can say that the results are not due to surface plasmons but interference in the multilayer stack for two different light polarizations.

It is important to note that the dips in reflectance spectra in the Kretschmann configuration correspond to absorption enhancement rather than SPP excitation [25]. Instead, *tracking the magnitude of the electric field at the surface, $|E|^2$, gives information on the SPP excitation.*

As an example, the mentioned dip for p-polarization at 43.6° originates from interference in the layered structure. This incidence angle could not be used for the polarization-controlled switching, as almost all the energy of the p-polarized pulse is absorbed in the top layer (Fig. 1 (d)).

The middle Si_3N_4 layer has a refractive index $n_{\text{Si}_3\text{N}_4} = 2.02$ greater than that of the SiO_2 prism $n_{\text{SiO}_2} = 1.45$, so it does not produce novel SPP modes. Therefore, taking into account the high losses in GdFeCo, one may use the ordinary dispersion of GdFeCo/air plasmons[15] $\beta_{\text{SPP}} = k_0 \sqrt{\epsilon_{\text{GdFeCo}} \epsilon_d / (\epsilon_{\text{GdFeCo}} + \epsilon_d)}$. As a zero approximation β_{SPP}^0 , one may take $\epsilon_d = \epsilon_{\text{air}} = 1$. In order to take into account the presence of the 10-nm-thick Si_3N_4 cover layer, one may use averaging of the refractive index:

$$\epsilon_d^{(1)} = \frac{\int_0^{w(\text{Si}_3\text{N}_4)} \epsilon_{\text{Si}_3\text{N}_4} \exp(-2\gamma_{\text{SPP}}^{(0)} z) dz + \int_{w(\text{Si}_3\text{N}_4)}^{\infty} \epsilon_{\text{air}} \exp(-2\gamma_{\text{SPP}}^{(0)} z) dz}{\int_0^{\infty} \exp(-2\gamma_{\text{SPP}}^{(0)} z) dz}$$

and with respect to the mode profile in zero-order approximation: $|e(z)|^2 = e_{z=0} \exp(-2\gamma_{\text{SPP}}^{(0)} z)$,

$$\gamma_{\text{SPP}}^{(0)} = \sqrt{(\beta_{\text{SPP}}^{(0)})^2 - k_0^2 \epsilon_{\text{air}}}$$

The panels below show that the calculated dispersion of SPPs $\beta_{\text{SPP}}(\omega)$ (shown by the light blue dashed line) is in good agreement with the position of the maximum of $|E|^2$ (shown by light blue circles), and is not related to the minimum of R (shown by violet circles). Also, we provide the ratio between $|E|^2$ for p- and s- polarized light, clearly indicating that **significant enhancement of $|E|^2$ for p-polarized light is only observed below the light line, and the plasmon frequency is in good agreement with the calculated dispersion of the SPPs $\omega(\beta_{\text{SPP}})$.**

In Fig. 1 (d) in the main text, we have introduced the curve $|e|^2(\theta)$ calculated for the bottom surface of the bottom GdFeCo layer, showing that the maximum of absorption does not correspond to the maximum of the electromagnetic field (as already discussed). The SPP resonance is very broad due to the absorption in GdFeCo that has huge imaginary part of the dielectric permittivity: $\epsilon = -5.91 + 19.17i$.

On the other hand, the revised version of Fig. 1 (d) shows that the angle of incidence 59° is in the vicinity of the SPP resonance (in contrast to 76° which corresponds to the maximum of the absorption). At the same time, the angle 59° experimentally provides optimum results in terms of the spatial profile of the incident pulse, since it corresponds to near-normal incidence on the prism face.

Generally, we used a set of incidence angles for experimental verification rather than a single angle (they are marked with black arrows in Fig. 1 (d)-(e)), and the obtained results are in excellent agreement with theoretical estimations shown in Fig. 1 (d). Supplemental Note 5 shows these results, experimentally confirming the key role of the SPPs in the observed selective switching of the bilayered GdFeCo structure.

2. In Fig. 2, the curve for the magneto-optical intensity is for the Fe sublattice whereas the arrows are for the magnetization of the Gd atoms. Moreover it is also stated that in one layer the Gd atom magnetization are parallel and in the other layer it is anti-parallel to the net magnetization. In my opinion, it is a little confusing for readers who might not be so familiar with the intricacies of rare-earth ferromagnetic films. If the authors agree, wouldn't it make more sense to at least change the arrows to reflect the net magnetization, since we are talking about switching magnetic layers?

The reviewer raises a good point, which we have considered in-depth. If we do indeed use arrows to show the net magnetization, we will have antiparallel arrows corresponding to the topmost and bottommost levels in the hysteresis loop (Fig. 2). We feel that this is visually distracting, since non-experts would intuitively expect to see the topmost / bottommost level of the hysteresis loop having up-up / down-down arrows. To therefore make the figure as clear as possible, we prefer to keep the current style used for the arrows. We hope the reviewer understands our point of view.

REVIEWERS' COMMENTS:

Reviewer #1 (Remarks to the Author):

Dear Authors,

I feel that the points raised in the previous round of review have been satisfactorily addressed. The only minor point, which is newly raised in the revised version by introducing the normalized variable "e" instead of the electric field strength variable "E" in the original version, is that the notation "arb. units" in Fig. 1(b) and (c) should be removed because currently $\text{abs}(e)^2$ is normalized definitely and is no longer in "arb. units".

I am pleased to recommend publishing the paper.

Song-Jin Im

Reviewer #3 (Remarks to the Author):

I am satisfied with the response provided by the authors. The manuscript in its current form can be published.

Response to reviewers

We thank the reviewers for recommending publication of our manuscript. Here, we address point-by-point all the comments provided by the reviewers. The text marked in blue corresponds to the comments from the reviewers, and the subsequent black text indicates our response. In the revised manuscript / supplementary material, the revisions are marked in red.

Reviewer #1 (Remarks to the Author):

Dear Authors,

I feel that the points raised in the previous round of review have been satisfactorily addressed.

The only minor point, which is newly raised in the revised version by introducing the normalized variable “e” instead of the electric field strength variable “E” in the original version, is that the notation “arb. units” in Fig. 1(b) and (c) should be removed because currently $\text{abs}(e)^2$ is normalized definitely and is no longer in “arb. units”.

I am pleased to recommend publishing the paper.

Song-Jin Im

We thank the reviewer for recommending publication. Thank you also for noticing the error with the label “arb. units”, which we have now corrected.

Reviewer #3 (Remarks to the Author):

I am satisfied with the response provided by the authors. The manuscript in its current form can be published.

We thank the reviewer for supporting publication of our paper.